# Cathepsins Trigger Cell Death and Regulate Radioresistance in Glioblastoma

**DOI:** 10.3390/cells11244108

**Published:** 2022-12-17

**Authors:** Xiaofeng Ding, Chen Zhang, Huajian Chen, Minghua Ren, Xiaodong Liu

**Affiliations:** 1School of Public Health and Management, Wenzhou Medical University, Wenzhou 325035, China; 2South Zhejiang Institute of Radiation Medicine and Nuclear Technology, Wenzhou 325800, China; 3Key Laboratory of Watershed Science and Health of Zhejiang Province, Wenzhou Medical University, Wenzhou 325035, China; 4Department of Urinary Surgery, The First Affiliated Hospital of Harbin Medical University, Harbin 150001, China

**Keywords:** glioblastoma, cathepsins, radiotherapy, radiosensitivity, cell death

## Abstract

Treatment of glioblastoma (GBM) remains very challenging, and it is particularly important to find sensitive and specific molecular targets. In this work, we reveal the relationship between the expression of cathepsins and radioresistance in GBM. We analyzed cathepsins (cathepsin B, cathepsin D, cathepsin L, and cathepsin Z/X), which are highly associated with the radioresistance of GBM by regulating different types of cell death. Cathepsins could be potential targets for GBM treatment.

## 1. Introduction

Glioblastoma (GBM) is extremely aggressive with a poor prognosis, and its exact etiology is unknown [1]. The survival and prognosis of GBM are related to malignancy and treatment sensitivity. The treatment for GBM consists of surgical resection combined with chemoradiotherapy. Radiotherapy (RT) remains a critical component of the therapy for GBM and other malignancies. Ionizing radiation directly or indirectly causes oxidative injury at the local or abscopal site [2] including DNA damage, cell cycle arrest, apoptosis, etc.

Cathepsins, mainly lysosomal proteases, may degrade proteins under acidic pH conditions. According to their catalytic amino acids, cathepsins are classified into cysteine proteases (cathepsin B, C, F, H, L, K, O, S, W, and Z (also known as CTSX)), serine proteases (cathepsin A and G), and aspartic amino acid proteases (cathepsins D and E). Structurally, these enzymes have not only rather short active site cleavage, including three well-binding substrate-binding subsites (S2, S1, and S1′), but also rather extensive binding regions (S4, S3, S2′ and, S3′). This geometry distinguishes them from other protease classes, such as serine and aspartic proteases, with six and eight substrate binding sites. Exopeptidases (cathepsins B, C, H, and X) compared to endopeptidases (such as cathepsins L, S, V, and F) have structural features that facilitate the binding of N- and C-terminal groups at the active site of a substrate in cracks [3]. The activities of cathepsins are tightly controlled at the transcriptional and post-transcriptional levels by various regulators under physiological conditions. First, a cathepsin is synthesized as a preproenzyme, followed by the removal of the pro-peptide (also a signal peptide) from the endoplasmic reticulum, glycosylation in the Golgi apparatus, transport to the lysosome, and activation by controlled protein hydrolysis [4]. Secondly, it is worth noting that the absence of one cathepsin can be compensated by the others in a complex protein regulatory network. For example, in transgenic mouse models of pancreatic duct adenocarcinoma, breast cancer cells, and myeloid-derived suppressor cells, reduced expression of cathepsin B was compensated by cathepsin X/Z in breast cancer cells [5]. Finally, many studies have suggested a causal relationship between cathepsins and GBM. For example, the expression of CTSB (an exopeptidase, acting as an exo- or endopeptidase) and its enzymatic activity are associated with high tumor aggressiveness and grade in GBM [6]. Likewise, cathepsin D (CTSD) remains a normal and major component of lysosomes in glioma cells compared to normal astrocytes. In addition, cathepsin S (CTSS) is unexpressed in normal glial, neuronal, or endothelial cells, but is expressed in GBM cells [7]. 

Cathepsins are involved in cell death, invasion, and metastasis [8,9]. Extracellular secreted cathepsins can alter the tumor microenvironment by degrading the extracellular matrix (ECM) and activating, processing, or degrading various growth factors, cytokines, and chemokines. Cathepsins also disrupt cell–cell adhesion molecules that contribute to tissue invasion and metastasis, and thus represent an important class of proteins that contribute to cancer progression and spread [10]. In addition, it has been shown that proteolytic enzymes, including lysosomal endopeptidases (cathepsins), are inseparable from extracellular matrix components, and essential for the invasiveness of GBM [11]. A previous study found that CTSL is involved in X-ray-induced invasion and migration of human GBM U251 cells, and inhibiting the expression of CTSL can weaken the invasive ability of U251 cells [12]. In addition, X-ray significantly increased the levels of CTSL in cytoplasmic and nuclear of U251 cells, and the inhibition of CTSL increased radiation-induced changes [13]. It is inferred that CTSL may be involved in regulating the radiosensitivity of GBM in vitro. 

From the above, our results will elucidate the roles of cathepsins in regulating the radioresistance of GBM and provide a theoretical basis for fulfilling GBM treatment strategies. 

## 2. Materials and Methods

The correlation analysis between cathepsins and GBM prognosis was obtained from GeneExpression Profiling Interactive Analysis (GEPIA). The protein–protein interaction (PPI) analysis was based on the STRING public database. The functional heterogeneity of cancer cells was explored by ScRNA-seq and distinct functional states of cancer cells were detected by CancerSEA Online. Venn Diagrams were drawn for interaction protein intersection analysis. GBM (GSE153982, HK-374 cells) radiotherapy data were obtained from the Gene Expression Omnibus database (GEO), and differential gene expression analysis was performed using the NetworkAnalyst online tool. The KEGG enrichment analysis was performed using the DAVID and KOBAS online tools. The survival curve was plotted by the Kaplan–Meier method, and the difference between groups was tested by log-rank. The co-expressed genes were analyzed by the Pearson method; *p* < 0.05 was considered statistically significant. The websites used in this study were as follows:

GEPIA: http://gepia.cancer-pku.cn, (accessed on 30 December 2021)

CancerSEA: a cancer single-cell state atlas, (accessed on 20 November 2022)

STRING: https://cn.string-db.org, (accessed on 30 December 2021)

GEO: https://www.ncbi.nlm.nih.gov/geo/,(accessed on 10 March 2022)

NetworkAnalyst: https://www.networkanalyst.ca, (accessed on 30 June 2022)

DAVID: https://david.ncifcrf.gov/, (accessed on 10 June 2022)

KOBAS: http://kobas.cbi.pku.edu.cn/, (accessed on 10 June 2022)

## 3. Results

### 3.1. Cathepsins and Clinical Prognosis in GBM 

The involvement of cathepsins in the progression of malignancies is frequently associated with aberrant proteolytic regulatory mechanisms. We analyzed the relationship between cathepsins and GBM based on the GEPIA database and found that CTSS, cathepsin C (CTSC), cathepsin K (CTSK), cathepsin O (CTSO), CTSB, CTSL, CTSZ/X, and CTSD were highly expressed in GBM (Figure 1), while only CTSB, CTSD, CTSL, and CTSZ/X (*p* < 0.05) were associated with overall survival (OS) (Figure 2). To better understand the role of cathepsins in GBM, the relationship between cathepsins and the different functional states of GBM was explored based on the CancerSEA database (CancerSEA: a cancer single-cell state atlas). The results showed that only CTSB and CTSZ/X were significantly associated with GBM functional status (Appendix A). Combined with the GEPIA analyses, we finally decided to explore the potential biological relationship between CTSB, CTSD, CTSL, CTSZ/X, and GBM.

### 3.2. Predicted PPIs of Cathepsins

To further investigate the specific mechanism by which cathepsins are involved in the regulation of the prognosis in GBM, we screened the PPIs of CTSB, CTSL, CTSZ/X (cysteine protease), and CTSD (aspartic protease) according to the public database STRING (https://cn.string-db.org, accessed on 30 December 2021) to figure out the potential downstream targets. (Figure 3; Appendix A, and then we performed an overlapping analysis of the PPIs of CTSB, CTSL, CTSZ/X, and CTSD by using Jvenn mapping. The results showed only 13 identical interacting proteins (Appendix A), verifying that although the two protease families CTSB, CTSL, and CTSZ/X (cysteine protease) and CTSD (aspartic protease) have similar structural elements and share the same proteolytic machinery, their conformations and catalytic activities, tissue and cellular distribution patterns, and physiological roles are different. 

### 3.3. Cathepsins and Cell Death

Cathepsins are lysosomal proteolytic enzymes and might perform distinct roles in cancer progression, including invasion and apoptosis [10,14,15]. Additionally, cathepsins may equally possess tumor-suppressive roles [16], depending on the cellular context. Therefore, it is important to perform in vivo analysis to understand the functions of cathepsins in the pathobiology of GBM [17]. We first analyzed the interacting proteins of the four OS-related cathepsins (Appendix A), secondly, we selected a set of cell death-related genes with a good correlation [18] (Appendix A), and finally we analyzed and screened specifically for cell death-related interacting proteins (Appendix A, Intersection of interacting proteins of CTSB, CTSD, CTSL, and CTSZ/X with cell death-related genes; Appendix A). Cell death-related genes were obtained from the GO (http://geneontology.org, accessed on 10 October 2022) and KEGG (https://www.kegg.jp, accessed on 10 October 2022) databases [18]. Cell death-related interacting proteins were obtained based on the intersection of cathepsin-related interacting proteins and cell death-related genes.

#### 3.3.1. CTSB and Cell Death

CTSB belongs to the MEROPS peptidase subfamily C1A, also known as thiol protease. It contains the propeptide-C1 domain and pept-C1 domain. The propeptide-C1 domain is found at the N-terminal of CTSB and CTSB -like peptidases. They are synthesized as inactive zymogens. Activation of the peptidases occurs with the removal of the propeptide. The Pept-C1 domain can catalyze the hydrolysis of peptide bonds in the polypeptide chain through the mechanism of the sulfhydryl group of the cysteine residue in the active center acting as a nucleophile. CTSB and its interacting proteins most likely act through peptide bond hydrolysis [19,20]. CTSB can be involved in the regulation of apoptosis, necroptosis, autophagy, pyroptosis, and ferroptosis by interacting proteins (Figure 4, Blue arrows). However, the exact mechanism remains to be explored. 

Firstly, two distinct pathways of apoptosis are involved here, the death receptor pathway and the mitochondrial pathway. The death receptor pathway is initiated by death receptors and related ligands on the cell surface, and CTSB leads to apoptosis by activating caspases 3, 7, and 8 to regulate intracellular structural and functional proteins. The mitochondrial pathway involves many conserved signaling proteins and depends on mitochondrial integrity. This process is regulated by the BCL-2 family consisting of pro-apoptotic (BAX, BAK, BAD, BID, PUMA, BIM, and PYCARD) and anti-apoptotic (BCL-2, BCL-xL, BCL-w, and Mcl-1) factors that tightly regulate the balance between protein activities. CTSB can act on anti-apoptotic or pro-apoptotic factors, directly or indirectly regulating the release of cytochrome c from mitochondria to regulate apoptosis. Previous studies have also indicated that CTSB downregulation can reduce TNF-α-mediated apoptosis and autophagy, and that CTSB, CTSH, CTSL, and CTSS can cleave canonical caspase substrates such as procaspase-1, -3, and -8 [21,22] and release the pro-apoptotic mitochondrial enzyme, cytochrome c [23], which activates caspases and apoptosis. EGFR as a predicted interacting protein of CTSB and its inhibition in a mouse lung tumor model showed that it can participate in apoptosis by activating BIM and PUMA [24]. 

Secondly, CTSB is also involved in the regulation of TNF signaling pathway-mediated necroptosis. TNF binds to its receptor TNFR1, which consists of the TNFR1-related death domain proteins TRADD, TRAF2, and RIPK1, and an inhibitor of apoptosis (cIAP1 or cIAP2). CTSB may affect RIP1 ubiquitination by cleaving cIAPs, thereby regulating necroptosis. The necroptotic process is dominated by stimulated poly (ADP-ribose) polymerase 1 (PARP-1), Ca^2+^-dependent calpain (Calpain), and the proapoptotic BCL-2 member BAX. Then, the truncated AIF (tAIF) is released from the mitochondria, thereby redistributing tAIF from the cytoplasm into the nuclear compartment, which in turn promotes chromatin disassembly and loss of cell viability. The anti-apoptotic protein BCL-2 inhibits this release. The roles of calpain, CTSB, and PARP in the nuclear translocation of AIF have been studied in the pathology of cerebral ischemia [25], and the activation of calpain and CTSB favors the release of AIF [26]. Studies have shown that in long-term liver injury and fibrosis, elevated levels of cytoplasmic CTSB can activate NOD-like receptors of the pyrin domain 3 (NLRP3) inflammasome [27]. Inhibition of Hsp90, an interacting protein of CTSB, attenuated TNF-α-induced necroptosis cell death by inhibiting the activation of the RIP1–RIP3–MLKL pathway [28]. 

Thirdly, CTSB can participate in the regulation of autophagy through interacting proteins in three stages (induction, nucleation, and elongation; Figure 4). The process of autophagy following nutrient or growth factor deprivation, activation of AMPK, and/or inhibition of mTOR leads to the activation of ULK, which phosphorylates Beclin1, leading to the activation of VPS34 and formation of phagocytes. CTSB may participate in this process by interacting with beclin1. Phagocyte nucleation occurs at the onset of the autophagy cascade through the phosphorylation of phosphatidylinositol 3-kinase class III (PI3KC3) complex I, which then interacts with PI3P to attract the ATG12~ATG5–ATG16L1 complex, which contributes to the addition of signature Protein LC3 (microtubule-associated protein light chain 3) to the extended isolation membrane. LC3-I is then converted to LC3-II, which is the fulcrum for phagocyte membrane expansion and closure. ATG5, 7, 12, and LC3, the key functional proteins of autophagy, are the interacting proteins of CTSB and may be cleaved by CTSB. Therefore, CTSB may play an important role in the regulation of autophagy nucleation and elongation. Additionally, the downregulation of CTSB resulted in a significant decrease in the mRNA of MAP1LC3-1 and ATG5 in granulosa cells [29]. In addition, CTSB downregulation resulted in a significant decrease in MAP1LC3-1 and ATG5 mRNA in granulosa cells, suggesting multiple regulatory mechanisms for both by CTSB. 

Fourthly, CTSB induces pyroptosis by cleaving the inflammasome complex (NLRP1/NLRP3/AIM2) and activating caspase1. CTSB may aggravate acute pancreatitis (AP) by activating the NLRP3 inflammasome and promoting caspase-1-induced pyroptosis [30]. In addition, NLRP4, one of the CTSB-interacting proteins, has been demonstrated to induce a GSDMD-dependent pyroptosis [31,32].

Finally, CTSB also regulates ferroptosis by cleaving CP (Ceruloplasmin). CP assists ferroportin (FPN) in exporting Fe^2+^, thereby further participating in the regulation of ferroptosis [33]. CTSB can also indirectly induce ferroptosis by increasing unstable cellular iron content through interacting proteins ATG5 and ATG7 [34]. Evidence of ferroptosis in diseases ranging from neurotrauma to cancer underscores the importance of its identification for clinical application, which is further reinforced when CTSB-knockout primary fibroblasts do not respond to various inducers of ferroptosis [35]

#### 3.3.2. CTSD and Cell Death

CTSD belongs to the family of aspartic proteases, different from the family of cysteine proteases, containing the A1-propeptide, TAXi-N (*Triticum aestivum* endoxylanase inhibitor-N-termini), and TAXi-C (*Triticum aestivum* endoxylanase inhibitor-C-termini) domains [36,37,38] (https://smart.embl.de/smart/, accessed on 30 December 2021). Most eukaryotic endopeptidases (Merops family A1) are synthesized with the signal peptides and propeptides. The propeptide contains two helices that block the active site cleft, specifically the Asp11 residue conserved in pepsin, which forms hydrogen bonds with the conserved Arg residue in the propeptide. This hydrogen bond stabilizes the propeptide conformation and may be responsible for triggering the conversion of pepsinogen to pepsin under acidic conditions [39,40]. The domains TAXi-N and TAXi-C are initially found in aspartic protease and xylanase inhibitors. The N- and C-termini of members of this family provide the catalytic pockets necessary for proteolysis. CTSD can exert a proteolytic function through the domains TAXi-N and TAXi-C. Increasing evidence shows that CTSD is actively involved in apoptosis, necroptosis, autophagy, and ferroptosis through interacting proteins (Figure 4, red arrows). 

Firstly, intracellular CTSD is a key mediator in the activation of pro- and anti-apoptotic proteins or nuclear proteins involved in apoptosis. It can trigger BAX insertion into the mitochondrial membrane directly or indirectly through Bid lysis [41,42], causing both the release of mitochondrial cytochrome c and activation of caspase 3, leading to apoptosis [43,44]. It can also regulate apoptosis by disrupting the balance between BH3/BCL-2 [44,45]. By analyzing the interacting proteins of CTSD, we hypothesized that it could act on IAP to deregulate the inhibition of caspase 3/9 and cleave pro-apoptosis-related factors (e.g., caspase 3/9, TNF, TP53, cytochrome c, AKT, etc.) to regulate apoptosis. 

Secondly, CTSD cleaves caspase 8 and dissociates it from mitochondria. Caspase 8 dissociation may lead to elevated levels of RIP-1 ubiquitination and the induction of necroptosis [46]. In addition, we proposed that CTSD could block TNF-mediated signaling as well as act on Calpain or Bid to regulate necroptosis. 

Thirdly, CTSD can regulate autophagy. CTSD is one of the major lysosomal proteases indispensable for the maintenance of cellular proteostasis by turning over substrates of endocytosis, phagocytosis, and autophagy. CTSD-deficient mouse models show lysosomal hypertrophy and impaired autophagic flux in the visceral and central nervous systems, and these phenotypes can be ameliorated by recombinant human CTSD [47]. Another study showed that CTSD deficiency or dual deficiency of CTSB and CTSL resulted in the severe blockage of autophagic flux, demonstrating that these proteases are essential for lysosomal proteolysis [48]. Some researchers have also proposed that when the expression of CTSD is significantly reduced, the levels of autophagy in the late stage are reduced accordingly [49]. Ferroptosis is an autophagic cell death process [34,50,51]. However, in the process of autophagy, establishing whether CTSD can regulate autophagy or ferroptosis by acting on autophagy-related proteins (Beclin1, ATG5, and ATG7) needs further experiments.

#### 3.3.3. CTSL and Cell Death

CTSL contains the domains Inhibitor-I29 and Pept-C1 [52,53] (https://smart.embl.de/smart/, accessed on 30 December 2021). Inhibitor-I29 is the cathepsin propeptide inhibitor domain. This domain is located at the N-terminus of some C1 peptidases and acts as a propeptide, which prevents substrates from entering the active site. CTSL can be activated by interaction with a second peptidase or by autocatalytic cleavage to remove the N-terminal inhibitory domain. Both CTSB and CTSL belong to the papain family of cysteine proteases and, in addition, because CTSL contains the peptidase-C1 structural domain, it also belongs to the cysteine peptidase family C1, subfamily C1A (papain family, CA family) [54]. CTSB and CTSL play an important role in apoptosis and necroptosis. However, there are differences between the interacting proteins of these two cathepsins and the specific mechanisms involved in cell death. 

CTSL is responsible for the degradation of multiple proteins and is known to be involved in neuronal apoptosis associated with abnormal cell cycles [55]. Cleavage of IAP by CTSL relieved the inhibition of caspase 3/9, and CTSL disrupted the balance between BH3/BCL-2 to regulate apoptosis. We propose a hypothesis based on the interacting proteins of CTSL: CTSL may participate in necroptosis either through interacting proteins or by activating apoptosis and then participating in necroptosis.

#### 3.3.4. CTSZ/X and Cell Death

CTSZ/X contains the domain of peptidase-C1, also belongs to cysteine peptidase family C1, subfamily C1A, exhibits carboxymonopeptidase and carboxydipeptidase activities, and is involved in maintaining homeostasis and immune cell function [56,57] (https://smart.embl.de/smart/, accessed on 30 December 2021). CTSZ/X is involved in cell death via interacting proteins (Figure 4, purple arrows). CTXZ/X plays a critical role in apoptosis, necroptosis, ferroptosis, and autophagy. 

It has been shown that CTSZ/X cleaves IAP to deregulate caspase 3/9 inhibition and disrupts the balance between BH3/BCL-2 to regulate apoptosis. CTSZ/X also mediates the EGFR/PI3K/AKT signaling pathway [24,58], P53/AKT signaling pathway [59], PARP/CAPN1 signaling pathway [60], and IGF2R/caspase3 signaling pathways [61] which are involved in apoptosis (Figure 4, purple arrow). Strong upregulation of CTSX/Z caused by *H. pylori* (*Helicobacter pylori*) infection has been reported to be strongly associated with gastric carcinogenesis, while knockdown of CTSX/Z leads to G1 phase arrest and apoptosis [62]. Knockdown of CTSX/Z expression or treatment with specific CTSX/Z inhibitors protected the pheochromocytoma cell line PC12 and neuroblastoma cell line SH-SY5Y from 6-hydroxydopamine (6-OHDA) toxicity and reduced apoptosis [63]. In H9c2 cardiomyocytes, hypoxia induces activation of CTSB-interacting protein IGF2R and activation of caspase 3, leading to apoptosis [61]. Additionally, CTSZ/X regulates necroptosis, probably by cleaving IAPs during necroptosis.

Finally, we used CTSZ/X and its interacting proteins to identify potential mechanisms that may be involved in autophagy, necroptosis, and ferroptosis (Figure 4, purple arrows), but no evidence of CTSZ/X involvement in autophagy, necroptosis, and ferroptosis has been reported. Whether CTSZ/X can be involved in cell death through cleavage of interacting proteins requires further confirmation.

### 3.4. Cathepsin-Mediated Radioresistance in GBM

RT induces changes in cathepsin expression, which might be associated with the radioresistance of GBM. To investigate the specific mechanism of radioresistance, we performed differential gene expression analysis using the GEO database and found that CTSB, CTSL, and CTSD increased in GBM after radiotherapy (GSE153982). We combined the changes in interacting proteins of CTSB, CTSL, and CTSD after radiotherapy (Appendix A) to infer the possibility that CTSB, CTSL, and CTSD are involved in apoptosis, necroptosis, and autophagy. The increase in CTSB, CTSL, and CTSD after radiotherapy first affects mitochondrial permeability, participates in apoptosis and necroptosis through the p53 signaling pathway, or directly participates in apoptosis through the mitochondrial pathway; secondly, mitochondria permeability through DNA damage or activation of the p53 signaling pathway affected by calpain is involved in necroptosis; finally, it is involved in necroptosis by mediating the classical pathway (TNF/RIP1/RIP3 signaling pathway) (Figure 5). We also performed a functional enrichment analysis using proteins that were differentially expressed after IR (Figure 6, Appendix A). KEGG enrichment analysis results revealed that the top six enriched pathways for differential genes were the cell cycle, p53 signaling pathway, DNA replication, nucleocytoplasmic transport, lysosome, etc. The p53 signaling pathway is involved in apoptosis and senescence and inhibits tumorigenesis by preventing the growth and division of damaged and potentially precancerous cells. Studies in model systems have shown that enhancing or reactivating p53 activity in cancer cells, either in combination with stress-inducing therapy or alone, can improve the efficiency of cancer therapy. The p53 signaling pathway is crucially meaningful in regulating the apoptosis of human GBM cells. Knockdown of CTSK could promote p53-dependent BAX upregulation to enhance oxaliplatin-induced apoptosis [64]. The expression of CTSL was significantly increased in p53-mutated human non-small cell lung cancer tissues, suggesting that there may also be some regulation of CTSL expression by p53 [65]. Further studies have suggested that it would be a brilliant idea to identify the p53–cathepsin axis as the canonical framework for cell death. As mentioned above, combined with the results of our differential gene analysis before and after RT, we believe that an increase in CTSB, CTSL, and CTSD after radiotherapy can directly or indirectly activate or inhibit the occurrence of apoptosis, necroptosis, or autophagy, and the specific mechanism requires further experiments to verify.

## 4. Discussion

GBM is the most common primary brain tumor in adults. Conventional treatment consists of surgical resection followed by 6 weeks of radiation therapy and concurrent temozolomide chemotherapy. Despite this treatment, the prognosis is poor, with a median survival of 16 months. Hypofractionation and stereotactic radiosurgery for radiation therapy dose intensification is a promising strategy that has been explored to address these challenges [66]. In this study, we found that cathepsin (CTSB, CTSL, and CTSD) expression is upregulated after IR treatment in GBM, and cathepsins can regulate cell death in various ways through their interacting proteins. Cathepsins are promising targets for the radiosensitization of GBM.

Cathepsins belong to a broad family of peptidases that are normally active primarily in endosomes and lysosomes and are involved in the lysosomal death pathway [67]. The main feature of the lysosomal death pathway is lysosomal membrane permeation (LMP). During LMP, lysosomal membrane integrity is lost, luminal contents such as cathepsins are released into the cytoplasm, and cathepsins cause damage to organelles and induce cell death such as apoptosis, pyroptosis, or necroptosis [68,69]. Further studies have shown that LMP is associated with radiosensitivity. Irradiation-induced lysosomal biogenesis leads to the release of more lysosomal hydrolysate into the cytoplasm, induces more cell death, and achieves radiosensitization [70]. The specific effect of cathepsins on cell death depends on whether their cleavage of the substrate is activating or inactivating. For example, in glioma stem cells, CTSK is responsible for the cleavage and inactivation of stromal-derived factor-1a, and this inactivation promotes stemness loss and increased sensitivity to chemoradiotherapy in glioma stem cells [71]. By contrast, CTSB processes BID into active tBID excessively, which induces the release of cytochrome C from the mitochondria. Second, we should also consider that cathepsins are affected by upstream signaling factors. For example, cervical cancer cells with knockdown of the cystine protease inhibitor SERPINB3 (squamous cell carcinoma antigen 1, SCCA1) are more sensitive to ionizing radiation, mainly due to lysosomal-like death caused by enhanced CTSL activity [72,73]. Thirdly, several studies on cathepsin-mediated radiosensitivity of GBM have been reported. For example, the knockdown of CTSB resulted in cell cycle arrest in the G0/G1 phase, where the efficiency of homologous recombination was impaired [74]. In addition, CTSD was found to be more significantly upregulated in radiation-resistant clones than in parental U251 cells, and the knockdown of CTSD improved the radiosensitivity of U251 cells. Subsequent studies found that CTSD modulates the radiosensitivity of GBM by affecting the fusion of autophagosomes and lysosomes [75]. Similarly, CTSL is involved in regulating the radiosensitivity of glioma cells by acting as an upstream regulator of NF-κB activation, and inhibition of CTSL sensitizes glioma cells to radiation [76]. In sum, it is important to understand the complete biology of cathepsins in GBM, including their impact on cell death, invasion, and the tumor microenvironment. This will improve the benefits of clinical treatments for GBM.

The field of cysteine cathepsins has undergone major changes in recent years, with CTSL, CTSB, and CTSD as possible targets for GBM therapy. For example, KGP94, a small molecule inhibitor of CTSL, has shown good therapeutic effects in vitro against the breast and prostate cell lines [77], and we speculate that inhibition of CTSL in GBM may likewise be an effective targeted therapeutic approach. Inhibition of CTSB has shown promising results in GBM studies. Caffeine reduces mRNA, protein expression, and activity of CTSB via Rho-associated protein kinase (ROCK), which in turn reduces glioma cell invasion and tumor growth in an orthotopic xenograft animal model via the FAK/ERK signaling pathway [78]. Furthermore, CTSB is shown to be a direct target of miR-140, and inhibition by miR-140 results in reduced temozolomide resistance and cell migration [79]. Since cathepsins are ubiquitous proteases and distributed in various tissues and organs, targeting them could cause corresponding side effects in different organs, such as lung fibrosis [80], morphea-like skin lesions [81,82], cardiovascular complications [83], etc. Therefore, targeting cathepsin presents significant challenges. It is important to fully understand the role of cathepsin in the physiological and pathological processes of various tissues and organs, and balance the clinical benefits and organ toxicity. The specific mechanism of cathepsins involved in cell death studied in this paper can provide ideas for designing small molecule drugs and minimizing side effects. It should also be noted that CTSL have tumor-promoting or tumor-suppressing functions in different preclinical models [84,85]. Targeting one of the cathepsins will cause the compensation of its diminished activity by the others and the impact on an illness would be reduced. The activities of other cathepsins will compensate for the reduced activity of the targeted cathepsin, which means that their activities will not diminish; e.g., the constitutive loss of CTSH can be partially functionally compensated by other cysteine- or aspartate-type proteases of type II pneumocytes, such as CTSC and CTSE [86]. CTSL was found to be upregulated in the thyroid glands of mice deficient in CTSK or CTSB, suggesting that cathepsin L compensates for cathepsin K deficiency [87]. Therefore, targeting one of the cathepsins may not achieve the desired inhibitory effect, but the compensatory mechanism of cathepsins may be a promising research direction, and targeting the sensors of the compensatory mechanism or rational use of pan-cysteine cathepsin inhibitors targeting multiple family members (pan-cathepsin inhibitor JPM-OEt) [88] may provide an idea for targeted therapy. Therefore, understanding the complete biological properties of these cathepsins will allow us to obtain the highest clinical benefit. There are still some limitations in our study, as cysteine cathepsins are expressed in inactive performance and their activation and activities are dependent on many factors, such as mRNA and protein expression, tumor epigenetics, etc. The study would have much higher relevance if protein levels would be included. At the same time, based on the results provided by ScRNA-seq, we also found that cathepsins have significant relationships with different functional states of GBM (especially CTSB and CTSZ/X). Finally, cathepsins, especially extracellular cathepsins, are now also considered suitable targets for targeted drug delivery. In addition, the application of cathepsins, especially CTSB, in some prodrugs and antibody–drug conjugates can successfully activate and release drugs [89]. This trend may even increase in the future, showing that cathepsin holds some promise for future drug discovery.

## 5. Conclusions

In conclusion, this paper elucidates the role of cathepsins in the regulation of cell death and cathepsins are potential targets for GBM radiosensitization.

## Figures and Tables

**Figure 1 cells-11-04108-f001:**
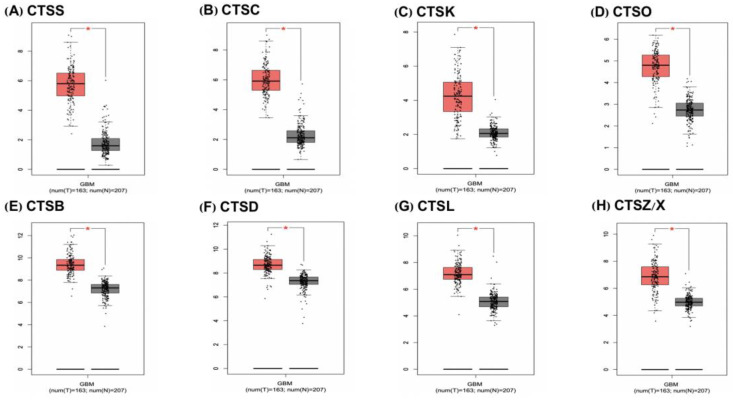
Cathepsin mRNA expression is upregulated in GBM patients. (**A**–**H**) Compared with normal tissues, CTSS, CTSC, CTSK, CTSO, CTSB, CTSD, CTSL, and CTSZ/X are highly expressed in tumors. Number of tumor samples (n = 163); number of tissue samples (n = 207). The red boxplot represents the tumor, and the grey boxplot represents the normal tissue. The data are presented as the mean ± SD, * *p* < 0.05.

**Figure 2 cells-11-04108-f002:**
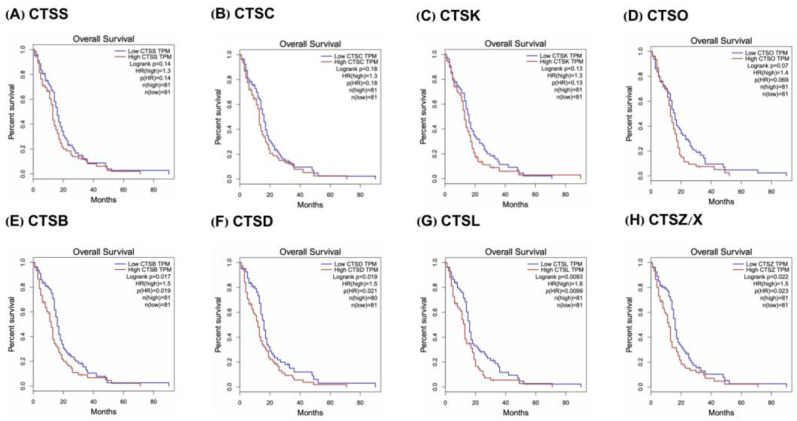
Cathepsin mRNA expression level in the OS of GBM patients. (**A**–**D**) The expressions of CTSS, CTSC, CTSK, and CTSO were not significantly associated with shortened overall survival in GBM patients (*p* > 0.05). (**E**–**H**) The expressions of CTSB, CTSD, CTSL, and CTSZ/X were significantly associated with shorter overall survival in GBM patients (*p* > 0.05). Number of tumor samples (n *=* 162).

**Figure 3 cells-11-04108-f003:**
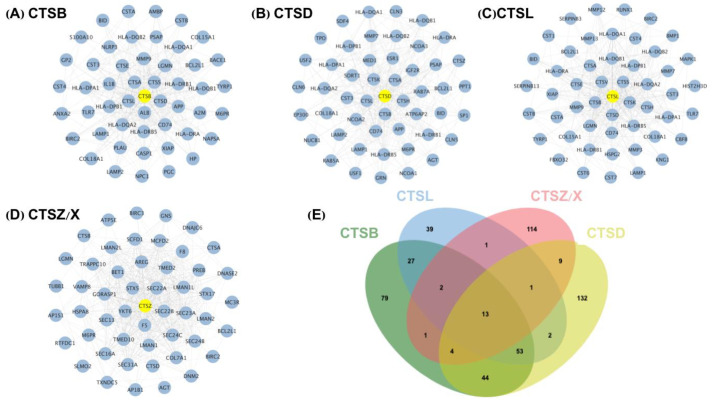
Top 50 PPIs of CTSB (**A**); CTSD (**B**); CTSL (**C**); CTSZ/X (**D**); Overlapping interacting proteins of four cathepsins (**E**).

**Figure 4 cells-11-04108-f004:**
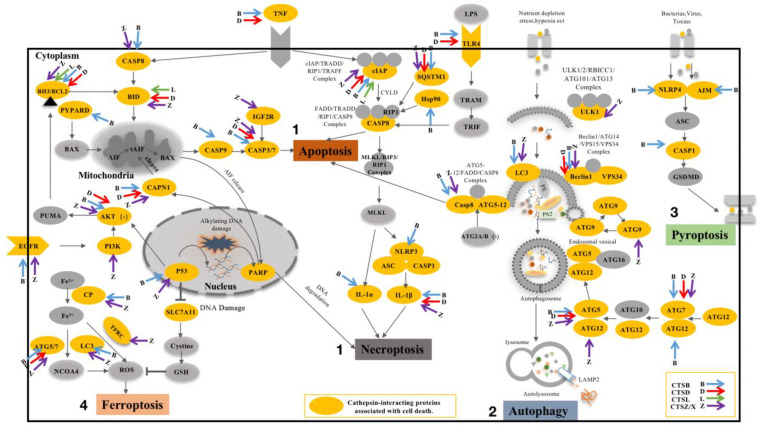
The involvement of CTSB/CTSD/CTSL/CTSZ/X-interacting proteins in cell death. Yellow circles indicate cathepsin-interacting proteins associated with cell death. Blue arrows represent the interacting proteins of CTSB involved in cell death; red arrows represent the interacting proteins of CTSD involved in cell death; green arrows represent the interacting proteins of CTSL involved in cell death; purple arrows represent the interacting proteins of CTSZ/X involved in cell death. 1. The process of apoptosis and necroptosis. 2. The process of autophagy. 3. Pyroptosis process. 4. Ferroptosis process.

**Figure 5 cells-11-04108-f005:**
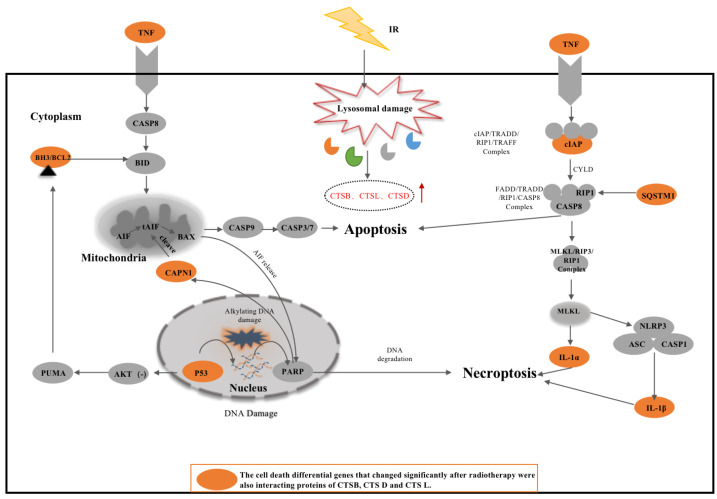
Differentially expressed CTSB, CTSL, and CTSD-interacting proteins after RT are involved in cell death. The orange circles represent the differentially expressed CTSB, CTSL, and CTSD-interacting proteins after RT.

**Figure 6 cells-11-04108-f006:**
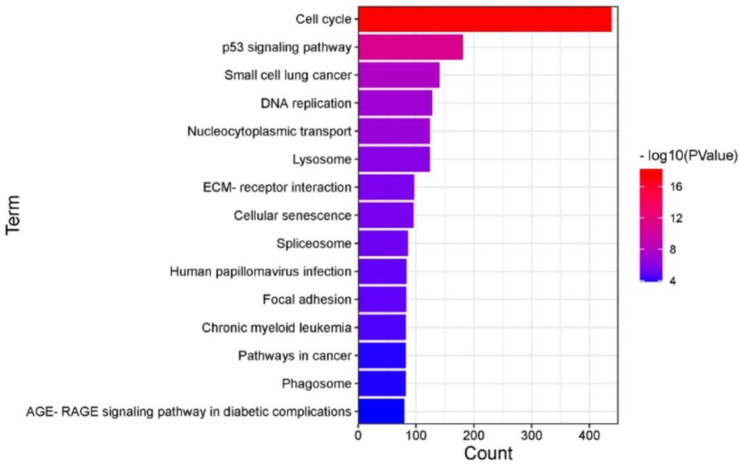
KEGG pathway enrichment analysis of differential genes after radiotherapy in GBM. In the GEO database, 12 samples (radiotherapy-treated in GBM cells) were analyzed. Differential genes after RT were analyzed using DAVID and KOBAS.

## Data Availability

GEO public database: https://www.ncbi.nlm.nih.gov/, accessed on 10 March 2022; String public database: https://cn.string-db.org/, accessed on 30 December 2021; GEPIA public database: http://gepia.cancer-pku.cn/, accessed on 30 December 2021; CancerSEA: a cancer single-cell state atlas.

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
