# Peer review of "Cathepsins Trigger Cell Death and Regulate Radioresistance in Glioblastoma"

_cells, 2022, doi:10.3390/cells11244108_

Round 1
Reviewer 1 Report
In the present study, Ding et al used several public datasets to demonstrate the relationship between glioblastoma and cathepsins. Their data displayed that cathepsins were highly expressed in glioblastoma. Expression of cathepsins is associated with apoptosis, necroptosis, pyroptosis, ferroptosis and autophagy. The current data indicates that cathepsins might be potential targets during the treatment of glioblastoma. However, there are some shortcomings in the current manuscript as below:
1. In Figure 1,2 and 9, the numbers of samples (n) should be indicated in the Figure legend.
2. In Figure 1, authors showed the cathepsins in normal and glioblastoma tissues. It would be better to show their expression in the different stages of glioblastoma.
3. In Figure 4,5 and 6, authors showed the interaction between the expression of cathepsins and cell death in glioblastoma. Authors should demonstrate the gene expression of cell death-related pathways in each figure.
4. Are there single cell RNA-Seq data available in glioblastoma? Authors can show the expression changes in different clusters from single cell RNA-Seq data and discuss the relationship between the expression of cathepsins and overall survival in glioblastoma.
Author Response
Dear Reviewers and Editors:
We feel great thanks for your professional review work on our article. As you are concerned, there are several problems that need to be addressed. According to your nice suggestions, we have made extensive corrections to our previous draft, the detailed corrections are listed below.
I have tried my best to improve my English writing ability. If reviewers still feel the need to improve their professional English writing skills, we plan to use English editing tools to revise the manuscript. (The reviewer comments are laid out below in italicized font and specific concerns have been numbered. Our response is given in normal font and changes/additions to the manuscript are given in the yellow text.)
- The review's comment:In Figure 1,2 and 9, the numbers of samples (n) should be indicated in the Figure legend.
The authors' answer: We have changed it correctly. In Figure 1,2:(Page 4 line 158)(Page 5 line 164).In Figure 9(Changed to Figure 6):In the GEO database, 12 samples(radiotherapy-treated in glioblastoma cells) were analyzed. Differential genes after radiotherapy were analyzed using an online website. (Page 12 line 473)
- The review's comment:In Figure 1, authors showed the cathepsins in normal and glioblastoma tissues. It would be better to show their expression in the different stages of glioblastoma.
The authors' answer: Many thanks to the reviewers for their suggestions. I immediately performed a literature search and database search and found that Glioblastoma is a rapidly growing grade 4 tumor. According to the National Comprehensive Cancer Network (NCCN) malignant tumor clinical practice guidelines. The expression of cathepsin was not shown in different stages of glioblastoma.
- The review's comment:In Figure 4,5 and 6, authors showed the interaction between the expression of cathepsins and cell death in glioblastoma. Authors should demonstrate the gene expression of cell death-related pathways in each figure.
The authors' answer: I have made major revisions to Figures 4, 5, 6 and 7 based on the comments of the second reviewer. Many thanks to the reviewers for their suggestions and guidance. We have specifically shown in the figure that interacting proteins as well as interacting proteins of different cathepsins may be involved in cell death pathways, using different colors.
- The review's comment:Are there single cell RNA-Seq data available in glioblastoma? Authors can show the expression changes in different clusters from single cell RNA-Seq data and discuss the relationship between the expression of cathepsins and overall survival in glioblastoma.
The authors' answer: We think this is a good proposal. Many thanks to the reviewers. We reviewed the relevant literature and found that single-cell RNA-Seq data utilized in glioblastoma generally explored the relationship between cellular immunity and the microenvironment. However, we still used the Cancer SEA database to analyze and explore CTSS, CTSC, CTSK, CTSO, CTSB, CTSD, CTSL, and CTSZ/X. We found that only CTSB and CTSZ/X of the eight cathepsins with significant and significant significance were associated with GBM prognosis (Fig. S1). Combined with the GEPIA database to analyze the prognostic relationship between cathepsin and GBM, we finally decided to explore the potential biological relationship between CTSB, CTSD and CTSL, and CTSZ/X and GBM.Among them,CancerSEA is the first dedicated database that aims to comprehensively decode distinct functional states of cancer cells at single-cell resolution. Providing a cancer single-cell functional state atlas, involving 14 functional states(Angiogenesis、Apoptosis、Cell Cycle、Differentiation、DNA damage、DNA repair、EMT、2Hypoxia、Invasion、Inflammation、Metastasis、Proliferation、Quiescence and Stemness) of 41,900 cancer single cells from 25 cancer types.

Author Response
Dear Reviewers and Editors:
We feel great thanks for your professional review work on our article. As you are concerned, there are many problems that need to be addressed. According to your nice suggestions, we have made extensive corrections to our previous draft, the detailed corrections are listed below.
I have tried my best to improve my English writing ability. If reviewers still feel the need to improve their professional English writing skills, we plan to use English editing tools to revise the manuscript. (The reviewer comments are laid out below in italicized font and specific concerns have been numbered. Our response is given in normal font and changes/additions to the manuscript are given in the yellow text.)
- The review's comment: lines 38-39: The authors wrote: “Proteolytic enzymes, including lysosomal endopeptidases, which are inseparable from extracellular matrix components, …” - It is not clear what the authors mean with this sentence. How can cathepsins be inseparable from extracellular components? Please explain more in detail or correct the sentence.
The authors' answer: We made the correct modification to the sentence: “Cathepsin secreted extracellular can alter the tumor microenvironment and degrade the extracellular matrix (ECM) as well as the activation, processing, or degradation of various growth factors, cytokines, and chemokines. Cathepsin also disrupts cell-cell adhesion molecules that contribute to tissue invasion and metastasis, and thus represents an important class of proteins that contribute to cancer progression and spread5.”( Page 1 line 38-42).
- The review's comment:lines 45-61: The authors announce the explanation of different types of cathepsin activity regulation (lines45-46) but then, with the exception of the first case, they list some random features of cathepsins.
The authors' answer: Our explanation of the regulation of the activity of different types of cathepsins is further supplemented in detail, including the active site of cathepsins as well as the cleavage site. ( Page 2line 50-56).
- The review's comment:Figure 2 caption : The authors should briefly explain (in Figure caption or in the text) what is on this Figure and what distinguish data regarding cathepsins B, D, L and X/Z from data regarding other cathepsins. One can see nomajor differences between results a to h.
The authors' answer: We describe it in detail under the annotations in Figure 2:(A-D) The expression of CTSS, CTSC, CTSK and CTSO was not significantly associated with shortened overall survival in GBM patients (P>0.05). (E-H) The expressions of CTSB, CTSD, CTSL and CTSZ/X were significantly associated with shorter overall survival in GBM patients (P > 0.05). ( Page 5 line 167-169)
- The review's comment:Supplemental Table S1 : This table is lacking the table head. The reader can guess that the authors wanted to show the interacting proteins of the 4 cathepsins. But in what order? The table head must be added. It would also benice, if beside the abbreviations, also the full name of proteins would be written. Or there should be a link to a webpage, where fullnames can be
The authors' answer: We have added table head to Supplemental Table S1 and attached web links so readers can find details of these interactions. (Page 5 line 178-179)
- The review's comment:ines 176-184: This paragraph is missing reference/s!
The authors' answer: We have added references to this paragraph. (Page 6 line 194-204)
- The review's comment:Figures 4- 7: I suggest a major reorganisation of Figures 4-7:
- Nucleus and mitohondria need to be marked.
- The label for a PPI is not well chosen, since “x” is usually used to mark an inhibitory effect.
- The data are not clearly separated between different types of cell death. I recommend using a colored underlay for a spcificcell death process.
The authors' answer: We have made significant changes to Figures 4-7, and we also think that generalizing in one graph can achieve good results. We explicitly labeled nuclei and mitochondria and altered the labeling of PPIs. We use yellow circles to represent cathepsin PPIs (PPIs associated with five cell deaths simultaneously), and arrows of different colors represent the targets of different cathepsins. We also thought about using different colored arrows to distinguish different cell death methods, but worried about confusion again. If the reviewers feel that different colored arrows are also needed to distinguish different cell death methods, we can change it again. (Page 8 line 272)
- The review's comment:line 293: the authors are using terms like TAXi- N and TAXi-C . Although I come from the cathepsins`field I am not familiar with this term.
The authors' answer: Our website provides explanations of terms such as TAXi-N and TAXi-C. References are also attached. (Page 8 line 327-336)
- The review's comment:lines 337-367; 408; 414-419: majority of the statements are lacking references
The authors' answer: We have attached citations to this:(Page 6 line 208-215)(Page 9 line 327-340)(Page 10 line 366-375)(Page 10 line 385-389).
- The review's comment:Figure 8 and its caption are very confusing . What is shown on the left and what on the right side of thefigure? The caption states that PPIs of cathepsins before and after radiotherapy. But it looks that the authors show only PPIs afterradiotherapy. Also, the caption states that the increased PPI is shown in red and then, that it is shown in blue. What is in blueand what is in red? Red dashed boxes are missing in this figure also. The authors must clearly state what they are showing on thefigure. The labels on the figure must correspond to the explained label in caption.
The authors' answer:We have also made a major revision to this image, taking into account the previous major revisions to the merging of the four images into a single image. The orange circles represent differentially cell death genes that were significantly altered after radiotherapy and were also interacting proteins of CTSB, CTSD, and CTSL. At the same time, we also screened the differential genes of cell death that were significantly changed after radiotherapy (the absolute value of log2FC was greater than 0.5). Specific genes are attached in Supplementary Table 5.(Page 12 line 469-471)
- The review's comment:Supplemental Table S4: What is this table showing? There is no Table title and no Table legend. There isalso no explanation of the data in the text.
The authors' answer: We have added table titles to the Supplementary Tables and Supplementary Table S4 has become Supplementary Table S6 due to major revisions. Supplementary Table S6 is a gene set obtained by differential gene analysis from data found in the GEO database. We screened a subset of genes that changed significantly after radiotherapy. (P<0.05)(Page 11 line 434-435)
- The review's comment:Conclusions - lines 546-549: The authors are promoting cysteine proteases (cathepsins L, B, D) as possible targets in glioblastoma therapy since chemotherapy causes side effects. The authors should discuss the followingproblem: cysteine cathepsins are ubiquitous proteases and targeting them would cause several problems. Next, in lines 549-554, the authors are talking about extracellular proteolysis being a network of interconnected mechanisms and that by targeting oneof them may have severe effect on several other proteolytic enzymes. It is more likely that targeting one of the cathepsins will cause the compensation of its dimished activity by the others and that the impact on illness would be reduced. The authors shouldcomment on this aspect also. Finally, in line 559 the authors stated, that their “disease prognostic data analysis is based on mRNAlevels”. As cysteine cathepsins are expressed in inactive proforms and their activation and activities are dependant on manyfactors, the study would have much higher relevance if protein levels would be included. As they are not, this should be commentedon.
The authors' answer: We first used the therapeutic efficacy of cathepsin inhibitors to explain that targeting one of these cathepsins results in diminished activity of the other cathepsins, thereby reducing the impact on disease. (Page 14 line 535-544)
Second, we use the case of CTSG to side-point that extracellular proteolysis is a network of interconnected mechanisms and targeting one of them may have serious consequences for several other proteolytic enzymes. (Page 14 line 546-550)
Finally, "Analysis of disease prognostic data based on mRNA levels". Since cysteine cathepsins are expressed in an inactive form and their activation and activity depend on many factors, the study would be more relevant if protein levels were included. However, after reading a lot of literature, most of the articles also used this method for subsequent experiments (doi: 10.1186/s12943-021-01383-x and doi: 10.1186/s12943-020-1143-7). We explain this briefly, and we can remove this section from the manuscript if the reviewer deems it inappropriate.
We also add the prospect of cathepsin inhibition and the broader role of cathepsins in drug activation and release in addition to their targeting role (Page 14 line 554-556)
13.The review's comment: In general, there are very little references from the Turk group in Slovenia (Jozef Stefan Institute) -they have worked on cathepsins B, L, D, Z/X and others for many decades, including on their involvement in apoptosis. The authors should review their references also and include them in their discussion.
The authors' answer: I am very grateful to the reviewers for the research on cathepsins in the Turk group in Slovenia (Jozef Stefan Institute) recommended by the reviewers, which is conducive to further enriching the content of this article and understanding cathepsins. We also add the corresponding literature in the corresponding part.
- The review's comment: The authors should refer to cathepsin X/Z throughout the text; sometimes they use only cathepsin Z.
The authors' answer: We have made uniform modifications to cathepsin X/Z.
- The review's comment:line 47: cathepsins are synthesized as preproenzymes and not proenzymes; please correct
The authors' answer: We have changed it correctly. (Page 2 line 59-60)
- The review's comment:line 48: in endoplasmatic reticulum prepeptid (also signal peptid) is removed; please change propeptides to prepeptides or signal peptides
The authors' answer: We have changed it correctly. (Page 2 line 59-60)
- The review's comment: line 89: in vitro should be in italic: in vitro
The authors' answer: We have changed it correctly. (Page 3 line 101,Page 9 line 319,Page 13 line 516,Page 14 line 537)
- The review's comment:Figures 1, 2 and 3: please use larger font, so that the data will be readable
The authors' answer: We have changed it correctly.
- The review's comment:lines 140 and 146/147: The authors are repeating the mistake, that cathepsins B, L and X/Z are aspartic proteases and that catepsin D is a cysteine protease. It is the other wayaround.
The authors' answer: We have changed it correctly. (Page 5 line 173-174,179-180)
- The review's comment:ines 257, 259; Figure 4 and Suppl. Table S1: The authors use the terms BECN1, Beclin-1 and BELIN-1: do they mean the same protein or different proteins? Please correct if these are all the same protein.
The authors' answer: We uniformly changed to Beclin 1.(Page 8 line 298-302)
- The review's comment:Figure 3E: lightly colored names of cathepsins cannot be seen. The authors should use black color.
The authors' answer: We have made the font larger and bolder for readers to see clearly. (Page6 line 191)
- The review's comment:line 168: in vivo should be in italic: in vivo
The authors' answer: We have changed it correctly. (Page6 line 198)
- The review's comment:It looks like the sections are not properly numbered: 3.3.1 (line 291) should be in fact 3.3.2; 3.3.1 (line 336) should be in fact 3.3.3 and 3.3.1 (line 400) should be in fact 3.3.4.
The authors' answer: We have changed it correctly. ( line 207,325,364,383)
- The review's comment:line 438 and Figure 8: the authors use the term caplain; they probably mean calpain
The authors' answer: We have changed calpain uniformly to CAPN1, for consistency with interacting protein names, we can change to calpain if not needed. (Page12 line 471)
- The review's comment: line 472: in vivo should be in italic: in vivo
The authors' answer: We have changed it correctly. (Page11 line 465)
- The review's comment:reference 47 (line 694) lacks “authors” - The cancer genome Atlas Research Network
The authors' answer: We have included "The cancer genome Atlas Research Network" as a database and provided a URL link. (Page3 line 109-112)

Round 2
Reviewer 1 Report
All of my comments have been addressed. No further comments.
Author Response
Dear Reviewers and Editors:
We feel great thanks for your professional review work on our article. I have finished revising the manuscript. We added a new author: Chen Zhang, who offers constructive suggestions for the revision of the article.
Kind regards,
Xiaofeng Ding
Author Response
Dear Reviewers and Editors:
We feel great thanks for your professional review work on our article. According to your suggestions, we have made the corrections one by one. In addition, we added a new author: Chen Zhang, who offers constructive suggestions for the revision of the article. (The changes/additions to the manuscript are given in the green text)
General comments - major
- The review's comment: The authors wrote: “To better understand the mysterious and complex roles of CTSS, CTSC, CTSK, CTSO, CTSB, CTSD and CTSL 196 and CTSZ/X expression in cancer, we explored the functional status in GBM based on the CancerSEA database,” and refer to the Figure S1. But the Figure S1 title (line 964) is “The CancerSEA tool explores the relationship between cathepsin expression and different functional states in tumors”. The authors must be strict - did they explore the functional status of cathepsins in GBM or in tumor cell in general?
Reply: According to your suggestion, we have corrected the inappropriate sentence in the revised manuscript. The new sentence is also shown below (Page 3 line 101-103): To better understand the role of cathepsins in GBM, the relationship between cathepsins and different functional states of GBM was explored based on the CancerSEA database". The title of Figure S1 has also been corrected to "The relationship between cathepsins and different functional states of GBM was explored based on the CancerSEA database".
- The review's comment: Please join Figures S2 and S3 together into 1 Figure. Please use larger font, as the numbers inside the circles can not be read.
Reply: We have joined Figures S2 and S3 into Figure S2 and used larger font for easier identification by the reader.
- The review's comment: lines 949-953: The reference 93 is incorrectly cited - this reference does not talk about cathepsin G inhibiting MMPs. There is no mentioning of MMPs in this reference.
Reply: We have deleted the incorrect reference and related sentence.
- The review's comment:Conclusion was much better written in the first version of the manuscript. But, the authors should improve it by addressing and commenting on the following issues more in detail (as I have mentioned before):
1) cysteine cathepsins are ubiquitous proteases and targeting them would cause several problems.
2) It is more likely that targeting one of the cathepsins will cause the compensation of its dimished activity by the others and that the impact on illness would be reduced. (The authors do not understand it correclty: the activites of other cathepsins will compensate for the reduced activity of the targeted cathepsin, that means that their activities will not diminish, as they stated in their answer to the Editor and Reviewers (I can not find this statement in the revised manuscript).
3) As cysteine cathepsins are expressed in inactive proforms and their activation and activities are dependant on many factors, the study would have much higher relevance if protein levels would be included.
The authors added new information to the Conclusion that would be better if included in the Discussion section.
2). Targeting one of the cathepsins will cause the compensation of its diminished activity by the others and that the impact on illness would be reduced. The activities of other cathepsins will compensate for the reduced activity of the targeted cathepsins, that means that their activities will not diminish e.g: the constitutive loss of CTSH can be partially functionally compensated by other cysteine- or aspartate-type proteases of type II pneumocytes, such as CTSC and CTSE[5].Besides, CTSL was found to be upregulated in the thyroid glands of mice deficient in CTSK or CTSB, suggesting that cathepsin L compensates for CTSK and CTSB deficiency[6]. Therefore, targeting one of the cathepsins may not achieve the desired inhibition effect, but the compensatory mechanism of cathepsins may be a promising research direction, and targeting the sensors of the compensatory mechanism or rational use of pan-cysteine cathepsin inhibitors targeting multiple family members (pan-cathepsin inhibitor JPM-OEt) [7] may provide an idea for the targeted therapy. (Page 12 line 419-430)
3). As cysteine cathepsins are expressed in inactive proforms and their activation and activities are dependant on many factors, such mRNA and protein expression, tumor epigenetics, etc[8,9]. According to your suggestion,we will further study both the protein expression, epigenetic phenotypes and functional activity of cathepsins to improve its higher relevance with GBM. (Page 12 line 432-436)
We have included a new information in the Discussion section. (Page 11, line 401-409;Page 12, line 436-441).
General comments - minor
- The review's comment:line 71: The authors should refer to cathepsin X/Z (CTS X/Z) throughout the text; sometimes they use only cathepsin Z or CTSX.
Reply: We have uniformly corrected to cathepsin Z/X (CTSZ/X).
- The review's comment:- line 72: There is 1 “protease” too many in the sentence.
Reply: It has been revised. (Page 2 line 46-48)
- The review's comment:- line 76: Cathepsin B is listed as an exopeptidase. It can act as an exo- or -endo peptidase. This information should be included.
Reply: It has been revised. (Page 2 line 51-54)
- The review's comment:- line 191: “Cathepsin Z” should be “Cathespin X/Z”.
Reply: We have corrected it.
- The review's comment:- line 193: OS means “overall survival” and not “over survival”.
Reply: It has been revised. (Page 3 line 101)
- The review's comment:- lines 230 and 236: The authors did not correct the mistake: cathepsins B, L and X/Z are cysteine proteases (not aspartic) and catepsin D is an aspartic protease (not cysteine).
Reply: It has been corrected. (Page 4 line 123-124)
- The review's comment:- lines 485, 486 and 582: Triticum aestivum and Helicobacter pylori (H. pylori) should be written in italics.
Reply: It has been corrected (Page 7 line 237,238) (Page 8 line 307)
- Bühling, F.; Röcken, C.; Brasch, F.; Hartig, R.; Yasuda, Y.; Saftig, P.; Brömme, D.; Welte, T. Pivotal role of cathepsin K in lung fibrosis. Am J Pathol 2004, 164, 2203-2216, doi:10.1016/s0002-9440(10)63777-7.
- Falgueyret, J.P.; Desmarais, S.; Oballa, R.; Black, W.C.; Cromlish, W.; Khougaz, K.; Lamontagne, S.; Massé, F.; Riendeau, D.; Toulmond, S.; et al. Lysosomotropism of basic cathepsin K inhibitors contributes to increased cellular potencies against off-target cathepsins and reduced functional selectivity. J Med Chem 2005, 48, 7535-7543, doi:10.1021/jm0504961.
- Desmarais, S.; Black, W.C.; Oballa, R.; Lamontagne, S.; Riendeau, D.; Tawa, P.; Duong, L.T.; Pickarski, M.; Percival, M.D. Effect of cathepsin k inhibitor basicity on in vivo off-target activities. Mol Pharmacol 2008, 73, 147-156, doi:10.1124/mol.107.039511.
- Mullard, A. Merck &Co. drops osteoporosis drug odanacatib. Nat Rev Drug Discov 2016, 15, 669, doi:10.1038/nrd.2016.207.
- Bühling, F.; Kouadio, M.; Chwieralski, C.E.; Kern, U.; Hohlfeld, J.M.; Klemm, N.; Friedrichs, N.; Roth, W.; Deussing, J.M.; Peters, C.; et al. Gene targeting of the cysteine peptidase cathepsin H impairs lung surfactant in mice. PLoS One 2011, 6, e26247, doi:10.1371/journal.pone.0026247.
- Friedrichs, B.; Tepel, C.; Reinheckel, T.; Deussing, J.; von Figura, K.; Herzog, V.; Peters, C.; Saftig, P.; Brix, K. Thyroid functions of mouse cathepsins B, K, and L. J Clin Invest 2003, 111, 1733-1745, doi:10.1172/jci15990.
- Joyce, J.A.; Baruch, A.; Chehade, K.; Meyer-Morse, N.; Giraudo, E.; Tsai, F.Y.; Greenbaum, D.C.; Hager, J.H.; Bogyo, M.; Hanahan, D. Cathepsin cysteine proteases are effectors of invasive growth and angiogenesis during multistage tumorigenesis. Cancer Cell 2004, 5, 443-453, doi:10.1016/s1535-6108(04)00111-4.
- Li, J.; Xie, H.; Ying, Y.; Chen, H.; Yan, H.; He, L.; Xu, M.; Xu, X.; Liang, Z.; Liu, B.; et al. YTHDF2 mediates the mRNA degradation of the tumor suppressors to induce AKT phosphorylation in N6-methyladenosine-dependent way in prostate cancer. Mol Cancer 2020, 19, 152, doi:10.1186/s12943-020-01267-6.
- Yang, X.; Zhang, S.; He, C.; Xue, P.; Zhang, L.; He, Z.; Zang, L.; Feng, B.; Sun, J.; Zheng, M. METTL14 suppresses proliferation and metastasis of colorectal cancer by down-regulating oncogenic long non-coding RNA XIST. Mol Cancer 2020, 19, 46, doi:10.1186/s12943-020-1146-4.
